# Patterns of long-term care utilization during the last five years of life among Swedish older adults with and without dementia

Atiqur SM-Rahman[1,2]*, Bettina Meinow[3,4], Lars-Christer Hydén[1,5], Susanne Kelfve[1,3]

**1** Department of Culture and Society (IKOS), Division Ageing and Social Change (ASC), Linkoping University, Linköping, Sweden, **2** Faculty of Health, School of Health Policy and Management, York University, York, Canada, **3** Department of Neurobiology, Karolinska Institutet and Stockholm University, Aging Research Center, Care Sciences and Society, Solna, Sweden, **4** Stockholm Gerontology Research Center, Stockholm, Sweden, **5** Center for Dementia Research (CEDER), Linkoping University, Linköping, Sweden

* atiqur.rahman@liu.se

**Data Availability Statement:** Data cannot be shared publicly because of specific privacy restrictions. Data are available from the Division Ageing and Social Change at Linköping University

## Abstract

### Aims

The aims of this study were to compare the patterns of long-term care (LTC) use (no care, homecare, residential care) among people with and without dementia aged 70+ in Sweden during their last five years of life and its association with sociodemographic factors (age, gender, education, cohabitation status) and time with a dementia diagnosis.

### Methods

This retrospective cohort study included all people who died in November 2019 aged 70 years and older (n = 6294) derived from several national registers. A multinomial logistic regression was conducted to identify which sociodemographic factors predicted the patterns of LTC use.

### Results

Results showed that the time with a dementia diagnosis and cohabitation status were important predictors that influence the patterns of LTC use during the last five years of life. Nearly three-quarters of people living with dementia (PlwD) used residential care during the last five years of life. PlwD were more likely to reside in residential care close to death. Women who lived alone, with or without dementia, used residential care to a higher degree compared to married or cohabiting women.

### Conclusions

Among people without a dementia diagnosis, as well as those who were newly diagnosed, it was common to have no LTC at all, or use LTC only for a brief period close to death. During the last five years of life, PlwD and those living alone more often entered LTC early and

Institutional Data Access / Requests for access to the data that support the findings of this study can be put to the Head of Division, Division Ageing and Social Change, Linköping University (LiU) [andreas.motelklingebiel@liu.se] who meet the criteria for access to confidential data and will be handled according to the relevant legislation.

**Funding:** This project has received funding from the European Union's Horizon 2020 research and innovation programme under the Marie Skłodowska-Curie grant agreement No 764632. The funders had no role in study design, data collection and analysis, decision to publish, or preparation of the manuscript.

**Competing interests:** The authors have declared that no competing interests exist.

used residential care for a longer time compared to people without dementia and people living alone, respectively.

## Background

Dementia diseases are progressive, and care needs among people living with dementia (PlwD) increase throughout the different stages of the disease. The need for long-term care (LTC) accelerates during the last years of life, when individuals with dementia increasingly need help with daily life activities [1,2]. Although studies have addressed care needs of PlwD [3,4] and their LTC use in some countries such as in Australia [5], in the UK [6], little is known about how PlwD use LTC during their last years of live in a Swedish context.

In Sweden, LTC (homecare and residential care) for older people are publicly financed, needs-based, and managed by the municipalities. Municipal need-assessors decide on social care needs based on an individual assessment. However, there are no uniform rules that specify the amount of help to which a person is entitled to, implying a high degree of autonomy for the municipalities. While the proportion of older adults aged 65+ who received homecare remained almost unchanged over the past decade, the provision of residential care decreased by 30% (e.g., from 5.4% in 2010 to 3.7% in 2020) [7]. The long prevailing policy of "aging in place" has entailed that individuals with complex care needs are increasingly cared for at home [8] and need assessments for access to residential care have become stricter [9]. Although most PlwD live in ordinary housing [3], cognitive impairment has been shown to be a major factor associated to residential care.

The patterns of LTC use and specifically the length of stay in residential care has mostly been described in previous cross-sectional studies based on population register data in Sweden [10], in Finland [11] and in Canada [12]. Swedish research addressing older people's use of LTC highlighted a decreasing length of stay in residential care [13]. Further, studies on use of LTC have mostly focused on the general population [14,15] and less is known about care use among PlwD. Recent studies showed that about three quarters of all users of LTC for older people had a dementia diagnosis [2] and nearly half of PlwD live in a residential care facility [2,16], while a quarter of them did not use any eldercare [4]. Yet, little is known about the differences in the use of LTC between PlwD and older adults in general and what type of LTC that PlwD use during their last years of life.

### Aim

Based on national register data comprising the whole Swedish population, the aim of this study was to (i) compare the patterns of LTC use (no care, homecare, residential care) among people with and without dementia aged 70+ in Sweden during their last 5 years of life and to (ii) assess the association of sociodemographic factors (age, gender, education, cohabitation status) and time with dementia diagnosis with patterns of LTC use.

## Methods

### Study design

This is a retrospective cohort study based on Swedish register data.

### Data sources

Data were derived from four different Swedish registers linking the Longitudinal Integration Database for Health Insurance and Labour Market Studies (LISA) with the National Cause of

Death Register (CDR), the National Patient Registers (NPR), and the Social Service Register (SSR). Information on socio-demographic factors and the level of education was extracted from the LISA, administered by Statistics Sweden. While the CDR database provides dementia diagnoses based on death certificates, data on dementia diagnoses before death came from the NPR (in-patient care as well as specialist care), both administered by the National Board of Health and Welfare (NBHW). The record of LTC services was retrieved from the SSR database, that gathers monthly data on granted and provided LTC, also administered by the NBHW.

## Study population

The study population included all older adults living in Sweden, who died in November 2019 aged 70 years and older. It is important to note that, although the number of deaths may differ between months throughout the year, it can be assumed that patterns of LTC did not differ depending on the month of death. As 65 years is the minimum age for being eligible for LTC and individuals were followed retrospectively for the last five years of life, only those aged 70 years and over at the time of death in 2019 were included (n = 6294). To analyze factors associated with the use of LTC we compared three groups: people who were never diagnosed with dementia, people diagnosed with dementia before death, and people diagnosed with dementia in the death certificate. The dementia diagnosis followed ICD10-codes (F00-F03 or G30-G32) using NPR (before death) and CDR (death certificate).

## Measures

**Outcome measures.** The primary outcome variable was the pattern of LTC during the last five years of life. Based on the type and the duration of LTC, the outcome variable was grouped into six types:

1. no LTC;

2. entered late, only homecare = received homecare for 2 years or less;

3. entered late, end in residential care = received only residential care for 2 years or less;

4. entered early, only homecare = received homecare for at least 3 of the last 5 years;

5. entered early, end in residential care = entered homecare during year 1–3 and end in residential care;

6. stayed at least 4–5 years in residential care.

**Covariates (Socio-demographic variables).** We considered the following covariates: *gender*, *age at death* grouped in six age classes- 70 to 74 years, 75 to 79 years, 80 to 85 years, 86 to 89 years, 90 to 95 years, and 95+ years; *education* (the highest level of education was categorized into three levels: compulsory, i.e., primary level up to grade 9; secondary, i.e., gymnasium up to grade 12; and tertiary, i.e., university level); *cohabitation status* in November 2018 (the latest available information) was classified as cohabiting (married or share household) or living alone; and *time with dementia diagnosis* was categorized as diagnosis postmortem, 1 year or less, 2–3 years, 4–6 years, and 7+ years before death. In the analyses people with missing information were treated as a separate category.

**Statistical analysis.** First, descriptive statistics were conducted to calculate frequency and percentage of the different patterns of LTC use and demographic characteristics of people with a dementia diagnosis, compared to the total cohort. Second, a multinomial logistic regression

was conducted to identify to what degree time with dementia and cohabitation status predicted the patterns of LTC use (presented as predicted proportions). STATA MP15 was used for all statistical analyses.

**Ethics approval.** Ethical approval for record-linkage of the Swedish register data was obtained from Linköping Regional Ethical Review Board (Dnr 2016/293-31).

## Results

Of all individuals aged 70+ who died during November 2019, 29% had a dementia diagnosis at time of death (Table 1). Compared to persons without a dementia diagnosis, those with a diagnosis were older (mean age 86.9, SD = 6.5 vs. mean age 83.6, SD = 8.0), and more often women (60%). At time of death, about three quarters (74%) of PlwD and two thirds (67%) of those without a dementia diagnosis were living alone. The educational level did not differ between people with and without dementia.

People without dementia more often had not used any LTC at time of death (39%) compared to PlwD (29%). On the contrary, PlwD more often lived in residential care (38%) at time of death, compared to those without dementia (23%). The proportion of those who only used

**Table 1. Description of the study population aged 70+ with (PlwD) and without a dementia diagnosis (PwithoutD).**

|  | PlwD | PwithoutD | Total |
|---|---|---|---|
| Gender | n = 1821 (29%) | n = 4473 (71%) | n = 6294 |
| Women | 60.1 | 50.2 | 53.0 |
| Men | 39.9 | 49.8 | 47.0 |
| Age at death (year) | | | |
| Mean (SD) | 86.9 (6.5) | 83.6 (8.0) | 84.6 (7.7) |
| 70–74 | 4.6 | 15.8 | 12.5 |
| 75–79 | 9.8 | 18.1 | 15.7 |
| 80–84 | 18.8 | 19.4 | 19.2 |
| 85–89 | 28.3 | 20.8 | 23.0 |
| 90–94 | 27.7 | 16.3 | 19.6 |
| 95+ | 10.8 | 9.6 | 9.9 |
| Education | | | |
| Compulsory | 49.0 | 47.2 | 47.7 |
| Secondary | 33.4 | 35.8 | 35.1 |
| Tertiary | 15.8 | 15.6 | 15.7 |
| Missing information | 1.8 | 1.3 | 1.5 |
| Cohabitation status at time of death | | | |
| Living alone | 74.4 | 67.3 | 69.4 |
| Cohabiting | 25.6 | 32.7 | 30.7 |
| Patterns of LTC use | | | |
| No LTC | 28.9 | 38.7 | |
| Entered late, only homecare | 19.6 | 23.5 | |
| Entered early, only homecare | 13.0 | 14.5 | |
| Entered late, end in residential care | 11.0 | 7.0 | |
| Entered early, end in residential care | 13.6 | 8.7 | |
| At least 4–5 years in residential care | 13.8 | 7.3 | |

Note: Entered late, only homecare = received homecare for 2 years or less; Entered late, end in residential care = received only residential care for 2 years or less; Entered early, only homecare = received homecare for at least 3 of the last 5 years; Entered early, end in residential care = entered homecare during year 1–3 and end in residential care; Stays at least 4–5 years in residential care.

**Table 2. Sociodemographic characteristics and time with dementia diagnosis by different types of LTC use during the last 5 years of life among all decedents aged 70+ (presented without brackets) and among the sub-sample of decedents aged 70+ with a dementia diagnosis (presented within brackets).**

| | No LTC n = 1821, 29.0% (n = 91, 5.0%) (Group 1) | Entered late, only HC n = 1232, 19.5% (n = 179, 9.8%) (Group 2) | Entered late, end in RC n = 821, 13.0% (n = 379, 20.8%) (Group 3) | Entered early, only HC n = 695, 11.0% (n = 174, 9.6%) (Group 4) | Entered early, end in RC n = 858, 13.6% (n = 457, 25.1%) (Group 5) | At least 4–5 years in RC n = 867, 13.7% (n = 541, 29.7%) (Group 6) | Total n = 6294, 100% (n = 1821, 100%) |
|---|---|---|---|---|---|---|---|
| *Time with dementia diagnosis* | | | | | | | |
| No dementia diagnosis | 95.0 (0) | 85.5 (0) | 45.5 (0) | 78.9 (0) | 46.7 (0) | 37.6 (0) | 71.0 (0) |
| *Time point of diagnosis* | | | | | | | |
| Postmortem | 1.2 (23.0) | 2.6 (17.9) | 13.0 (24.0) | 3.6 (17.2) | 13.9 (26.0) | 15.3 (24.6) | 6.7 (23.4) |
| 1 year or less before death | 2.0 (40.7) | 6.6 (45.8) | 15.1 (27.7) | 7.3 (34.5) | 9.8 (18.4) | 5.5 (8.9) | 6.6 (22.8) |
| 2–3 years before death | 0.8 (15.4) | 3.6 (25.1) | 15.7 (28.7) | 4.0 (18.4) | 12.6 (23.6) | 7.0 (11.3) | 5.8 (20.3) |
| 4–6 years before death | 0.8 (15.4) | 1.0 (7.3) | 7.3 (13.5) | 4.3 (20.7) | 12.3 (23.2) | 16.3 (26.0) | 5.7 (19.8) |
| 7+ years before death | 0.3 (5.5) | 0.5 (3.9) | 3.3 (6.0) | 2.0 (9.2) | 4.7 (8.7) | 18.2 (29.2) | 4.0 (13.7) |
| *Gender* | | | | | | | |
| Women | 40.0 (41.8) | 47.4 (40.8) | 50.6 (47.5) | 61.4 (62.6) | 65.7 (67.2) | 70.0 (71.5) | 53.0 (60.0) |
| Men | 60.0 (58.2) | 52.6 (59.2) | 49.4 (52.5) | 38.6 (37.4) | 34.3 (32.8) | 30.0 (28.5) | 47.0 (39.9) |
| *Age at death* | | | | | | | |
| 70–74 | 26.0 (9.9) | 13.0 (5.6) | 5.3 (6.3) | 6.7 (4.6) | 3.7 (3.5) | 3.8 (2.9) | 12.5 (4.6) |
| 75–79 | 26.1 (19.8) | 16.7 (13.9) | 15.1 (16.0) | 10.5 (7.5) | 5.6 (5.5) | 8.0 (6.8) | 15.7 (9.8) |
| 80–84 | 22.1 (20.8) | 22.5 (23.5) | 22.4 (22.9) | 16.6 (20.1) | 14.2 (17.0) | 13.6 (15.2) | 19.2 (18.8) |
| 85–89 | 16.6 (29.7) | 27.3 (32.4) | 25.7 (29.5) | 24.9 (24.1) | 24.1 (25.2) | 24.8 (29.8) | 23.0 (28.3) |
| 90–94 | 7.4 (16.5) | 15.3 (20.7) | 23.3 (20.3) | 27.9 (30.5) | 33.0 (36.3) | 27.3 (29.0) | 19.6 (27.7) |
| 95+ | 1.9 (3.3) | 5.1 (3.9) | 8.0 (4.7) | 13.5 (13.2) | 19.3 (12.5) | 22.5 (16.3) | 10.0 (10.8) |
| *Education* | | | | | | | |
| Compulsory | 42.5 (43.3) | 46.2 (45.4) | 50.0 (47.7) | 52.7 (51.5) | 53.3 (52.3) | 54.1 (51.5) | 48.4 (49.9) |
| Secondary | 39.2 (41.1) | 37.5 (39.6) | 34.5 (33.8) | 32.5 (29.9) | 32.2 (32.2) | 32.8 (33.8) | 35.6 (33.9) |
| Tertiary | 18.2 (15.6) | 16.3 (14.9) | 15.5 (18.5) | 14.8 (18.6) | 14.4 (15.4) | 13.0 (14.6) | 16.0 (16.1) |
| *Cohabitation status* | | | | | | | |
| Cohabiting | 47.4 (52.7) | 39.5 (46.4) | 34.7 (42.5) | 15.9 (22.4) | 15.0 (17.3) | 9.0 (10.4) | 30.7 (25.6) |
| Living alone | 52.5 (47.3) | 60.5 (53.6) | 65.3 (57.5) | 84.1 (77.6) | 85.0 (82.7) | 91.0 (89.6) | 69.3 (74.4) |

Note: Entered late, only homecare (HC) = received homecare for 2 years or less; Entered late, end in residential care (RC) = received only residential care for 2 years or less; Entered early, only HC = received homecare for at least 3 of the last 5 years; Entered early, end in RC = entered homecare during year 1–3 and end in residential care; Stays at least 4–5 years in RC.

homecare during their last 5 years of life was similar among people with (33%) and without (38%) dementia. The patterns of LTC during the last 5 years of life among people with and without dementia and its association with sociodemographic characteristics and time with dementia diagnosis are presented in Table 2.

Almost one in three decedents had not used any LTC at time of death (Group 1, 29%) or they used LTC for a short period, and only homecare (Group 2, 20%). These two groups consisted mainly of people without dementia diagnosis (Group 1, 95%; Group 2, 86%). Men (60%) and relatively young decedents (70–84 years; 74%) were overrepresented among those who did not use any LTC (Group 1). A majority (61%) of those who only used homecare for a shorter period at the end of life (Group 2) were living alone. Few people were found in group 4, i.e., they used only homecare for three years or more out of the last 5 years of life (11%). People in this group mainly had no dementia diagnosis (79%), were women (61%), and lived alone (84%).

40% of the older decedents lived in residential care at time of death. Some entered late and used LTC for two years at most (Group 3, 13%) and some entered early and used LTC for at least 4–5 years, progressing from homecare to residential care (Group 5, 14%). In these two groups people with dementia diagnosis were slightly overrepresented (Group 3, 55%; Group 5, 53%). Decedents who had entered LTC early and progressed from homecare to residential care (Group 5) were predominantly living alone (85%). Two thirds (66%) were women, and a majority (55%) died at a relatively old age (85–94). A dementia diagnosis was most common among decedents who had spent at least 4 years in residential care (62%; Group 6). This group mainly consisted of women (70%) and people living alone (91%) and death at an age of 95 + was more common (23%), compared to all other groups (2–19%).

Among PlwD (numbers are presented within brackets in Table 2), very few had not used any LTC (Group 1, 5%) or had entered LTC late and had only used homecare (Group 2, 10%) at time of death. Compared to other groups, the proportion of newly diagnosed (<1 year) individuals was higher in these two groups (Group 1, 41%; Group 2, 46%), as was the proportion of relatively young decedents (Group 1, 30% <80 years, Group 2, 20%) and cohabiting people (Group 1, 53%; Group 2, 46%). A majority of them were men (Group 1, 58%; Group 2, 59%). Newly diagnosed decedents were also relatively common among PlwD who only used homecare during at least three of the last five years of live (Group 4, 35%).

Almost three quarters of PlwD lived in residential care at time of death (n = 1377, 76%) and it was more common among PlwD to enter LTC early (n = 998, 55%). While newly diagnosed people with dementia (<1 year) were relatively common (28%) among those who had entered LTC late and lived in residential care at time of death (Group 3), those who had entered LTC early (Group 5) had more commonly lived with a dementia diagnosis for a longer time (2–7 + years, 56%). These two groups mainly consisted of people who died at a relatively old age (85–94 years: Group 3, 50%; Group 5, 62%), and a majority lived alone (Group 3, 58%; Group 5, 83%). Almost a third of PlwD spent 4 years or more time in residential care (Group 6, 30%). This group mainly consisted of people who had lived a longer time with a dementia diagnosis (4+ years, 55%), women (72%), older people (85+ years, 75%) and people living alone (90%).

Summarizing Table 2, results show that those who used no LTC or who used only homecare for 2 years or less during their last 5 years of life were mainly people who had no dementia diagnosis at time of death. Those who did not use any or little LTC were more commonly men and younger decedents. In contrast, among those who used residential care, a majority had a dementia diagnosis, and the proportion increased the longer people had stayed in residential care. The vast majority of those who had lived in residential care longest were women and nearly one in four died at an age of 95+ years.

Table 2 indicates that patterns of LTC utilization were associated to socio-demographic characteristics, such as age, gender, and cohabitation status. Moreover, the time an individual had lived with a dementia diagnosis also played a crucial role for LTC trajectories. In Table 3, we used multinomial logistic regression models to estimate the predicted proportion of patterns of LTC utilization, by time with dementia and sociodemographic characteristics (age, sex, and cohabitation status). The level of education was not included in the analysis since it showed no significant impact on the patterns of LTC utilization. In this table the row % are designed to add to 100%.

In line with Table 2, the regression analysis (Table 3) shows that the predicted proportion of people who used no LTC was higher for people without dementia (36%) compared to PlwD (between 3–11%). The utilization of only homecare at time of death was determined by the duration of dementia diagnosis. For example, the predicted proportion of using only homecare was higher among people newly diagnosed with dementia compared to those with longer time with dementia (e.g., 1 year or less, 20%; 7+ years, 4%). Again, the multivariable analyses

**Table 3. Predicted proportion of LTC utilization for people aged 70+ with and without a dementia diagnosis by cohabitation status and time with dementia diagnosis.**

| | No LTC (Group 1) | Entered late, only HC (Group 2) | Entered late, end in RC (Group 3) | Entered early, only HC (Group 4) | Entered early, end in RC (Group 5) | At least 4–5 years in RC (Group 6) | Total |
|---|---|---|---|---|---|---|---|
| | Predicted prop. | Predicted prop. | Predicted prop. | Predicted prop. | Predicted prop. | Predicted prop. | |
| *Time with dementia diagnosis* | | | | | | | |
| No dementia | 36.0 | 23.6 | 7.3 | 15.4 | 9.7 | 7.9 | 100 |
| Diagnosed Postmortem | 10.2 | 10.4 | 24.5 | 6.7 | 23.0 | 25.4 | 100 |
| 1 year or less | 11.0 | 20.3 | 24.6 | 13.8 | 19.0 | 11.3 | 100 |
| 2–3 years | 5.1 | 13.8 | 30.4 | 8.4 | 27.0 | 15.4 | 100 |
| 4–6 years | 4.7 | 3.9 (n.s.) | 14.7 | 10.2 | 28.5 | 38.0 | 100 |
| 7+ years | 3.3 | 3.6 (n.s.) | 10.9 | 6.7 | 15.4 | 60.1 | 100 |
| *Cohabitation status* | | | | | | | |
| Cohabiting | 37.4 | 24.8 | 13.9 | 8.3 | 9.7 | 6.0 | 100 |
| Living alone | 24.7 | 18.0 | 10.4 | 15.3 | 15.2 | 16.4 | 100 |

Note: Results are based on one multinomial regression model including the total cohort (n = 6294), adjusting for age and gender. n.s = not significant, $p > 0.05$ (the total number of people with longer time with dementia was very few).

represent similar results as shown in Table 2, i.e., among PlwD, a majority of those who only used homecare had a dementia diagnosis for a shorter time. In general, the predicted proportion decedents in residential care were higher among people who had lived a longer time with dementia (e.g., 7+ years, 60%) compared to those who had lived a shorter time with dementia (e.g., 1 year or less between 19–25%). Residential care was also more common among those who were living alone.

## Discussion

Our findings particularly highlight the time with dementia diagnosis and cohabitation status as important predictors for patterns of LTC use during the last 5 years of life. Results from regression analyses also showed that the effect of these two predictors remained almost unchanged after adjustment for age at death and gender. The majority of people who used no LTC or used LTC only for a short period close to death, had no dementia diagnosis. We want to discuss five issues regarding the results.

First, several reasons could explain the differences in utilization of LTC between people with and without dementia: people without a dementia diagnosis may have less care needs or may die at younger ages. In line with previous studies [17,18] our results indicate that older adults without a dementia diagnosis may live an independent life until a few years before death. PlwD often have a much more protracted illness process, often involving the need for LTC a longer period compared to people without dementia. In line with our findings, an USA-based study found that higher levels of LTC utilization among PlwD were associated with disease severity [19].

Second, the results showed that nearly three quarters of PlwD used residential care during the last five years of life, similar to findings in an earlier study [20]. A possible explanation could be that PlwD require extensive care and continuous supervision during the last years of life, which makes homecare insufficient. Even an extensive provision of homecare with several visits around the clock entails that people are alone in their homes for around 20 hours a day–which would not meet the need of PlwD.

Third, our result showed that a substantial proportion of PlwD is more likely to live in residential care during the last years of life. It is, thus, possible that they die there. This assumption corresponds to a previous Swedish study where the authors showed that 89% of older adults with dementia died in a residential care facility [21]. This reflects the Swedish strong residential care establishment where many of these care facilities have special units for PlwD. Similar findings have been shown in several previous studies from the USA [22], and Ireland [23].

Fourth, our result showed that women, with or without dementia diagnosis, used residential care to a higher degree compared to men. This finding corresponds to previous studies [3,20]. The difference may be explained by women's higher life expectancy, longer period of dependency, and lower remarriage rates [24,25]. Our study findings based on the Swedish context are similar to those found women's greater use of residential care in studies from Switzerland [26] and Germany [27].

Finally, we did not find any significant effect of education on the patterns of LTC utilization. The lack of relationship between education and patterns of LTC use may be a result of Sweden's, in international perspective, comprehensive, universal and long established LTC system, that is largely financed by local taxes and available for all inhabitants aged $\geq$ 65. Individual income-related user fees are low, with a cap of currently ~225$ per month, covering 4–5% of the actual costs. Family or household economic resources are not considered. Our results are in line with previous studies that found only minor educational differences in LTC use, and only among women, after adjustment for cohabitation status and age at death. Accordingly, women with tertiary education used slightly less LTC compared to women with only primary education [28,29].

While there has been a greater focus on the use of LTC in general, individual patterns of LTC have rarely been investigated in previous literature. This study adds to existing knowledge by including longitudinal data during the last 5 years of life and included both people with and without dementia diagnosis. Although we have information about dementia diagnosis provided by hospital clinics, a main limitation of this study is the lack of data about dementia diagnosis provided by general practitioners in healthcare centers. Moreover, the time with dementia diagnosis may be underestimated since symptoms often develop gradually. Despite these limitations, the register data used in this study are of high quality (completeness: 97%; specificity: 99%) [30]. The main strength of our study is the nationwide coverage that includes all Swedish decedents from one specific month without dropout [31].

## Conclusion

To conclude, people without a dementia diagnosis, as well as those who were newly diagnosed, commonly did not use any LTC, or they used LTC only for a brief period close to death. PlwD who were living alone more often entered LTC early and used residential care for a longer time compared to those without dementia. As the number of beds in residential care have been reduced by about one third during the past two decades, people with dementia have increasingly been prioritized for access to the available beds [32]. Accordingly, cognitive impairment has, apart from limitations in basic activities of daily living, been found to be the main factor associated with being granted residential care. Since the number of PlwD is increasing in Sweden [33] these results indicate an increasing future demand for residential care.

## Author Contributions

**Conceptualization:** Atiqur SM-Rahman, Bettina Meinow, Lars-Christer Hydén, Susanne Kelfve.

**Data curation:** Atiqur SM-Rahman, Susanne Kelfve.

**Formal analysis:** Atiqur SM-Rahman, Bettina Meinow, Susanne Kelfve.

**Funding acquisition:** Lars-Christer Hydén.

**Methodology:** Atiqur SM-Rahman, Bettina Meinow, Lars-Christer Hydén, Susanne Kelfve.

**Software:** Atiqur SM-Rahman.

**Supervision:** Bettina Meinow, Lars-Christer Hydén, Susanne Kelfve.

**Visualization:** Atiqur SM-Rahman.

**Writing – original draft:** Atiqur SM-Rahman, Susanne Kelfve.

**Writing – review & editing:** Atiqur SM-Rahman, Bettina Meinow, Lars-Christer Hydén, Susanne Kelfve.

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
