## [Decision Letter · Decision Letter 0]

18 Jul 2023

PONE-D-23-14659Patterns of long-term care utilization during the last five years of life among Swedish older adults with and without dementiaPLOS ONE

Dear Dr. sm-Rahman,

Thank you for submitting your manuscript to PLOS ONE. After careful consideration, we feel that it has merit but does not fully meet PLOS ONE’s publication criteria as it currently stands. Therefore, we invite you to submit a revised version of the manuscript that addresses the points raised during the review process.

We look forward to receiving your revised manuscript.

Kind regards,

Kalkidan Hassen Abate, PhD

Academic Editor

PLOS ONE

“This project has received funding from the European Union’s Horizon 2020 research and innovation programme under the Marie Skłodowska-Curie grant agreement No 764632.”

Additional Editor Comments:

Dear Authors

As you can see from the comments of our reviewers, there are some basic problems with your reporting clarity and descriptions as a scientific paper. Since we believe that the value of the data and the conclusions drawn are worthy of publication, please complete the revised manuscript by addressing the reviewers' comments one by one.

Regards

Reviewers' comments:

Reviewer's Responses to Questions

**Comments to the Author**

1. Is the manuscript technically sound, and do the data support the conclusions?

Reviewer #1: Yes

Reviewer #2: Yes

2. Has the statistical analysis been performed appropriately and rigorously? 

Reviewer #1: Yes

Reviewer #2: Yes

3. Have the authors made all data underlying the findings in their manuscript fully available?

Reviewer #1: No

Reviewer #2: No

4. Is the manuscript presented in an intelligible fashion and written in standard English?

Reviewer #1: Yes

Reviewer #2: Yes

5. Review Comments to the Author

Reviewer #1: Thank you for the opportunity to review this interesting paper which seeks to compare patterns of long-term care (LTC) use at end of life for people with and without dementia in Sweden. Understanding patterns of care and drivers of care use can help inform planning and service delivery and so is an important topic.

1) Scope and aims: The data study uses high quality large-scale data from linkage of various Swedish registers. The analysis is mostly descriptive in nature and generally appears sound in relation to addressing the aims of the paper. There were only a limited number of socio-demographic factors investigated which limits the scope of the second aim of the paper which was to assess the association of sociodemographic factors and time with dementia on patterns of LTC use.

2) Background (first paragraph): Your last sentence notes that little is known about how PlwD use LTC during their last years of life. You could perhaps be more specific about this as to whether this is in a Swedish context or internationally. There are other studies that have looked at this question internationally (eg. https://pubmed.ncbi.nlm.nih.gov/33270824/)

3) Study population (page 5): This is well described but perhaps you could comment on whether selecting deaths occurring in just one month of the year is likely to be representative of all deaths in the year or whether this could bias the findings in any way?

4) Outcome measures (page 5): Suggestion only, but the description of the 6 outcome groups may be more readily interpretable in a figure or table showing the patient journey through care across the five years.

5) Socio-demographic variables (page 5): cohabitation status – could you please clarify throughout as to when this was measured? Was this at the beginning of the five-year period? At death?

6) Statistical analysis (page 6): There was insufficient detail re. the multinomial logistic regression model – please describe which variables were included in the model – was a separate model run for each variable? Was this run for the whole cohort or only for people living with dementia?

7) Results: Overall the results were comprehensively described. However, I found it confusing to switch between a comparison of people living with dementia compared to people without dementia (table 1) versus people living with dementia compared to the whole cohort (table 2). Table 2 in particular was confusing with the dementia cohort presented in brackets – is there a clearer way to display the results for the two cohorts?

8) Table 3: As mentioned at point 6 it was difficult to understand how this model was built and which cohort is being examined. I am assuming that it was the whole cohort and dementia and time with dementia diagnosis are combined into one variable? If so, this needs explaining a little further in the methods. Was it one model with four variables (age, sex, time with dementia + co-habitation)? When was co-habitation measured? Did you consider examining the interaction between dementia and co-habitation?

9) Discussion: I would have been interested to see some discussion about the lack of relationship between education and patterns of service use. I would have thought that socio-economic status (to which education is often correlated) could plausibly be a driver of LTC use and wonder what others have found in this area? Are the services provided in Sweden accessed in a more equitable fashion? Did you look at education in your model adjusted for age and gender?

Reviewer #2: Abstract. Do not talk about the "margins" command in the abstract. This is specific to Stata and the general reader would not know what this refers to.

Abstract. Conclusions last sentence. This is confusing: those living alone should be compared to those who do not live alone. Those with dementia should be compared to those without dementia. The final sentence of the Conclusions combines these comparisons and is therefore hard to follow.

Background. Is a reduction of the provision of residential care from 5.4% to 4.9% over 6 years really "drastic"? I would suggest using more measured language here or no emotive adjectives. You can perhaps just report the numbers as they are.

Statistical analysis. Do not refer to the "margins" command. I presume you are using Stata but you haven't told the reader this. I recommend describing the technique instead of the software command.

Table 2. Update row heading to "Time point of diagnosis"

Table 3. In the table headings/titles/footnote state the type of model used. Make it clear that in this table the row %s are designed to add to 100%. In this table what does a statistically significant value mean? How would be interpret the significant results.

Last line of the Results is an incomplete sentence...... "of Residential care was also.............". Please correct this.

6. PLOS authors have the option to publish the peer review history of their article (what does this mean?). If published, this will include your full peer review and any attached files.

Reviewer #1: No

Reviewer #2: **Yes: **Michael Waller

---

## [Author Response · Author response to Decision Letter 0]

22 Sep 2023

All the changes have made in the revised version of the manuscript are YELLOW marked in the document.

Response to Reviewers

Reviewer #1: Thank you for the opportunity to review this interesting paper which seeks to compare patterns of long-term care (LTC) use at end of life for people with and without dementia in Sweden. Understanding patterns of care and drivers of care use can help inform planning and service delivery and so is an important topic.

1) Scope and aims: The data study uses high quality large-scale data from linkage of various Swedish registers. The analysis is mostly descriptive in nature and generally appears sound in relation to addressing the aims of the paper. There were only a limited number of socio-demographic factors investigated which limits the scope of the second aim of the paper which was to assess the association of sociodemographic factors and time with dementia on patterns of LTC use.

Response: Thank you for this comment. We agree and will consider it for further research on the dataset. For this paper, we clarified the aim by specifying the sociodemographic factors included.

2) Background (first paragraph): Your last sentence notes that little is known about how PlwD use LTC during their last years of life. You could perhaps be more specific about this as to whether this is in a Swedish context or internationally. There are other studies that have looked at this question internationally (eg. https://pubmed.ncbi.nlm.nih.gov/33270824/)

Response: We revised the sentence.

3) Study population (page 5): This is well described but perhaps you could comment on whether selecting deaths occurring in just one month of the year is likely to be representative of all deaths in the year or whether this could bias the findings in any way?

Response: We revised the sentence accordingly.

4) Outcome measures (page 5): Suggestion only, but the description of the 6 outcome groups may be more readily interpretable in a figure or table showing the patient journey through care across the five years.

Response: Thank you for this suggestion. In the revised manuscript, we now converted the description into 6 bullet points and hope that this will improve readability. 

5) Socio-demographic variables (page 5): cohabitation status – could you please clarify throughout as to when this was measured? Was this at the beginning of the five-year period? At death?

Response: We used the latest available information on cohabitation status, which was from November 2018, and this information is now added in the manuscript.

6) Statistical analysis (page 6): There was insufficient detail re. the multinomial logistic regression model – please describe which variables were included in the model – was a separate model run for each variable? Was this run for the whole cohort or only for people living with dementia?

Response: We have now clarified (both in the method section and as Table descriptions) which variables that were included in the analyses and that the model was run for the whole cohort.

7) Results: Overall the results were comprehensively described. However, I found it confusing to switch between a comparison of people living with dementia compared to people without dementia (table 1) versus people living with dementia compared to the whole cohort (table 2). Table 2 in particular was confusing with the dementia cohort presented in brackets – is there a clearer way to display the results for the two cohorts?

Response: While Table 1 gives a description of the cohorts with and without dementia, our purpose with Table 2 was to give a picture of the distribution of sociodemographic factors and time with dementia within each “LTC pattern group” in total and for the sub-population of people with dementia. This could of course have been done differently, but a comparison between people with and without dementia would unable us from see the difference in the distribution of time with dementia in the “LTC pattern groups” for the total population, a result that we were interested in. 

We also agree that Table 2 is data intensive, thus, difficult to decipher. We tried several structures to display the results, but in the end, we found the current table structure as the most convenient way to display the results, both for the total population and for people living with dementia. To help the reader, we tried to clarify the structure of Table 2 by revising the Table title, which now more clearly state that it is the total population and the sub-population of people with dementia that is displayed.

8) Table 3: As mentioned at point 6 it was difficult to understand how this model was built and which cohort is being examined. I am assuming that it was the whole cohort and dementia and time with dementia diagnosis are combined into one variable? If so, this needs explaining a little further in the methods. Was it one model with four variables (age, sex, time with dementia + co-habitation)? When was co-habitation measured? Did you consider examining the interaction between dementia and co-habitation?

Response: We added the variables that were included in the model that formed the basis for table 3 and clarified in the method section when we measured cohabitation. We did try the interaction between dementia and cohabitation, but did not include it in the model since the interactions was not significant and did not improve the model fit.

9) Discussion: I would have been interested to see some discussion about the lack of relationship between education and patterns of service use. I would have thought that socio-economic status (to which education is often correlated) could plausibly be a driver of LTC use and wonder what others have found in this area? Are the services provided in Sweden accessed in a more equitable fashion? Did you look at education in your model adjusted for age and gender?

Response: Yes, we agree that this is an interesting point. We also tried a model where we included education, but it was not significant and did not improve the model fit and was hence excluded from the final model. We have now added more discussion about this result and also clarified that the Swedish elder care system is provided according to need, that is, accessible to all people 65 years and older after a needs assessment and is used by all socioeconomic strata. The fee is relatively low and also adjusted to the individual persons economic resources. 

 

Reviewer #2: Abstract. Do not talk about the "margins" command in the abstract. This is specific to Stata and the general reader would not know what this refers to.

Response: We agree and revised the sentence accordingly.

Abstract. Conclusions last sentence. This is confusing: those living alone should be compared to those who do not live alone. Those with dementia should be compared to those without dementia. The final sentence of the Conclusions combines these comparisons and is therefore hard to follow.

Response: Thank you for this observation. We have now revised the sentence.

Background. Is a reduction of the provision of residential care from 5.4% to 4.9% over 6 years really "drastic"? I would suggest using more measured language here or no emotive adjectives. You can perhaps just report the numbers as they are.

Response: We revised the sentence accordingly.

Statistical analysis. Do not refer to the "margins" command. I presume you are using Stata but you haven't told the reader this. I recommend describing the technique instead of the software command.

Response: We also revised this sentence.

Table 2. Update row heading to "Time point of diagnosis"

Response: We updated the heading.

Table 3. In the table headings/titles/footnote state the type of model used. Make it clear that in this table the row %s are designed to add to 100%. In this table what does a statistically significant value mean? How would be interpret the significant results.

Response: Thank you for this suggestion. We have now added information about the model in the footnote, as well as added the suggested sentence mentioning row % and added a "total" column in the table. Significance in this table refers to statistically significant coefficient in the multinomial regression model. 

Last line of the Results is an incomplete sentence...... "of Residential care was also.............". Please correct this.

Response: We have revised the sentence.

---

## [Editor Report · Decision Letter 1]

10 Oct 2023

Patterns of long-term care utilization during the last five years of life among Swedish older adults with and without dementia

PONE-D-23-14659R1

Dear Dr. sm-Rahman,

We’re pleased to inform you that your manuscript has been judged scientifically suitable for publication and will be formally accepted for publication once it meets all outstanding technical requirements.

Kind regards,

Kalkidan Hassen Abate, PhD

Academic Editor

PLOS ONE
---

## [Editor Report · Acceptance letter]

16 Oct 2023

PONE-D-23-14659R1 

Patterns of long-term care utilization during the last five years of life among Swedish older adults with and without dementia 

Dear Dr. SM-Rahman:

I'm pleased to inform you that your manuscript has been deemed suitable for publication in PLOS ONE. Congratulations! Your manuscript is now with our production department. 

Kind regards, 

on behalf of

Dr. Kalkidan Hassen Abate 

Academic Editor

PLOS ONE